# Effect of Nitrogen Nutrition and Planting Date on the Yield and Physicochemical Parameters of Flowering Chinese Cabbage

Wenping Liu [1], Małgorzata Muzolf-Panek [2] and Tomasz Kleiber [1,*]

[1] Laboratory of Plant Nutrition, Department of Plant Physiology, Faculty of Agronomy, Horticulture and Bioengineering, Poznan University of Life Sciences, Zgorzelecka 4, 60-198 Poznan, Poland

[2] Department of Food Quality and Safety Management, Poznan University of Life Sciences, 60-624 Poznan, Poland

* Correspondence: tkleiber@up.poznan.pl

**Abstract:** The nitrogen requirements of flowering Chinese cabbage are unknown. Our study aimed at investigating the effect of varied nitrogen nutrition levels (including 50, 70, 90, 110, and 130 mg N per $dm^3$, described as N-50 to N-130, respectively) on the yield of flowering Chinese cabbage (*Brassica campestris* L. ssp. *chinensis* var. *utilis* Tsen et Lee) grown in two varied soilless cultivation systems (substrate and hydroponic) and seasons (spring and autumn). We confirmed that the intensity of the nitrogen nutrition modified the yield of plants within a range of 50 to 90 mg N; the yields were increased; however, the higher N concentrations were not effective. In both cultivation systems, the content of K, Ca, and Fe in leaves was higher in autumn compared to spring. Nitrogen nutrition improved the weight of plants—an effect that varied depending on the system of cultivation—and increased the phenolic content. N-90 was the optimal level of nitrogen nutrition. More intensive N-nutrition did not significantly modify plant yields or phenolic content. We Concluded that N fertilisation might be an effective tool to obtain plants with high bioactive compound content.

**Keywords:** antioxidant activity; *Brassica campestris* L. ssp. *chinensis* var. *utilis* Tsen et Lee; chlorophyll content; colour; macroelements; microelements; nitrogen; phenolic content; yield

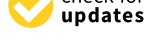



## 1. Introduction

Flowering Chinese cabbage (*Brassica campestris* L. ssp. *chinensis* var. utilis Tsen et Lee), also called "Caixin" in Mandarin Chinese or "Choy sum" in the Cantonese dialect, originated in China and then spread into other Asian countries such as Japan and Vietnam [1]. The edible parts of flowering Chinese cabbage are the leaves and flowering stem. Flowering Chinese cabbage belongs to the *Brassica* genus, which is not only a valuable source of glucosinolates, polyphenols, and vitamin C, but it is also rich in amino acids and other chemical compounds [2]. This vegetable has a high ability to accumulate selenium [3], which could be a beneficial element for consumers. However, there are still limited studies on the effect of nutrition on the yield and biochemical parameters of flowering Chinese cabbage grown in European conditions.

Nitrogen (N) fertiliser plays an important role in enhancing the yields of plants by affecting photosynthetic rates, growth rates, and plant productivity in most ecosystems [4–6]. Not only can it increase the volume of yields, but it also modifies the quality of the plants. This is because N nutrition is correlated with the biochemical and physiological functions of plants and is an essential constituent of protein, nucleic acids, chlorophylls, and growth hormones [4,5]. Insufficient nitrogen application might decrease leaf N concentration, chlorophyll content, and photosynthetic ability [6–8]. However, if the nitrogen application exceeds the ability of plants to use it, it depresses antioxidant activity, inhibits growth and other physiological process, and finally could be lethal for the plants [6,9,10]. Therefore, the determination of the optimal nitrogen application is one of the key factors to improve the yield and some biochemical properties of flowering Chinese cabbage.

The aim of this study was to understand the optimal nutritional requirements for Chinese cabbage and to achieve a yield that could obtain the optimal quality of this vegetable for consumers. Our hypothesis was nitrogen nutrition would improve plant yields independent of the planting date and system of cultivation. Thus, in this study, four independent experiments were conducted, including spring and autumn seasons as well as hydroponic and substrate cultivation systems in each season. The investigations covered the effect of nitrogen levels on yields and biochemical properties such as macro- and microelement content, total polyphenolic content (TPC), total flavonoid content (TFC), and the antioxidant activity of flowering Chinese cabbage.

## 2. Materials and Methods

The four vegetation experiments were conducted in 2018 in an unheated glasshouse (sector with an area of 272 $m^2$) located at the Experimental Station in Marcelin at the Poznań University of Life Sciences (Poland) on plants grown in pots with a substrate (I) (a mixture of loamy sand and peat) and in a hydroponic system with recirculation of a nutrient solution (II) (The experiments were conducted with the purple variety of flowering Chinese cabbage (*Brassica campestris* L. ssp. *chinensis* var. *utilis* Tsen et Lee) (Hubei Wuhan Hongshan Caitai Cultivation Center). The first period was the spring season (April–May), while the second one was autumn (September–October).

All the experiments were conducted in a completely randomised design with eight replicates in the hydroponic system experiment and ten replicates in the pot cultivation (a replicate was one single plant). The greenhouse was equipped with a climate control system. Both experiments checked the following nitrogen nutrition (in mg $N \cdot dm^{-3}$, respectively, for the nutrient solution and substrate): 50, 70, 90, 110, and 130 (presented as N-50, N-70 and N-90, N-110, and N-130, respectively). The nitrogen source was $NH_4NO_3$ (34% N).

Vegetation experiments were conducted using peat moss that was limed to a pH (in $H_2O$) of 6.00–6.50; doses of dolomite were established on the basis of the neutralization curve. The contents of available forms of nutrients in the peat moss substrate after it had been deacidified were as follows (in mg·dm$^{-3}$): N-NH$_4$ 35, N-NO$_3$ traces, P 20, K 18, Ca 1500, Mg 164, S-SO$_4$ 25, Fe 19.8, Zn 1.8, Mn 2.7, Cu 0.4, B 0.5, Na 18, and Cl 29 with EC 0.49 mS·cm$^{-1}$ at pH ($H_2O$) 6.00. At 14 days after liming, the other macro- and micronutrients were applied.

The pot experiments (I) were conducted on plants grown in yellow plastic pots (5 dm$^3$) filled with a mixture of mineral soil (loamy sand; bulk density of 1.60 g·cm$^{-3}$, total porosity 39%) and peat (*v/v/*1/1). Plants were grown in mixed substrate with the following chemical composition (mg·dm$^{-3}$): P—150, K—200, Ca—1500, Mg—200, Fe—75, Mn—25, Zn—20, Cu—10, and pH 6.0–6.5. The hydroponic experiments (II) were conducted on plants grown in a special hydroponic module with recirculation of the nutrient solution (NS). The nutrient solution for fertigation had the following chemical composition (mg dm$^{-3}$):, P-PO$_4$—50, K—200, Ca—120, Mg—60, S-SO$_4$—95, Fe—1.20, Mn—0.5, Zn—0.19, Cu—0.01, and B—0.011 with EC (electrolytic conductivity)—2.20 mS cm$^{-1}$.

The seedlings were prepared 2.5 weeks before the experiments. To prepare the homogenous plant material, the seeds were sown to rockwool fingers saturated with the stand nutrient solution. At the phase of 3–4 leaf seedlings, in the case of pot cultivation, they were transplanted into pots filled with a mixture of mineral soil and peat. In the hydroponic system, the seedlings were transplanted into rockwool blocks hydrated with the nutrient solution (Grodan, 100 × 100 × 65 mm). Ten days later, the plants were transplanted into the growing hydroponic system. During the cultivation, the plants were fertigated according to the requirements. The view of plants are shown in Figures 1 and 2.

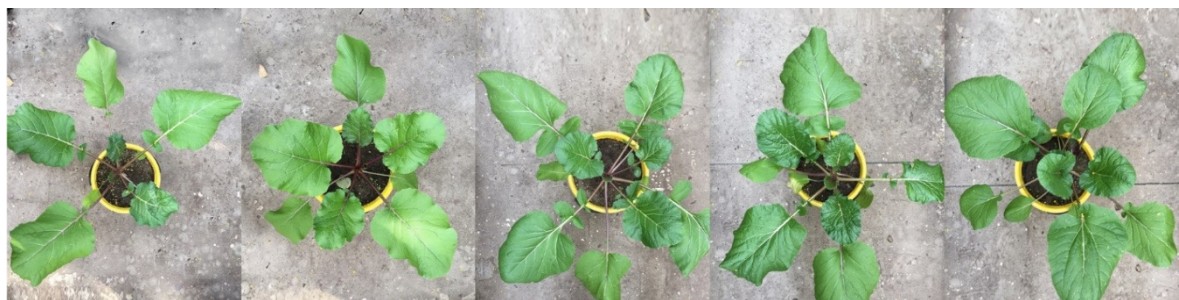

Substrate

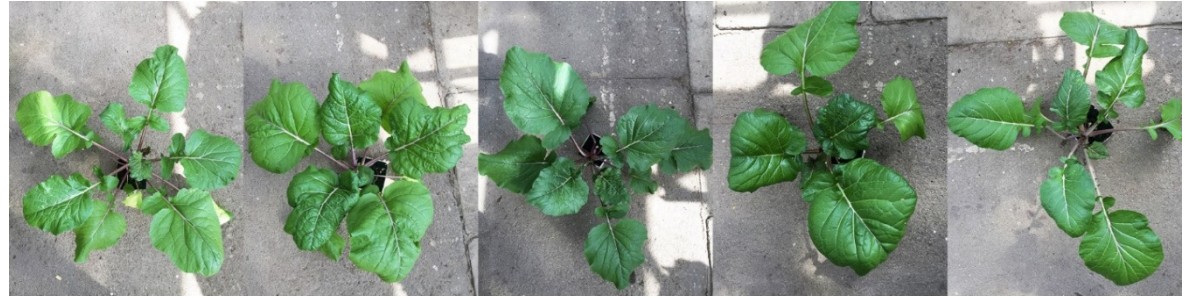

Hydroponic

**Figure 1.** Spring view of plants (from left: N = 50, 70. 90, 110, and 130).

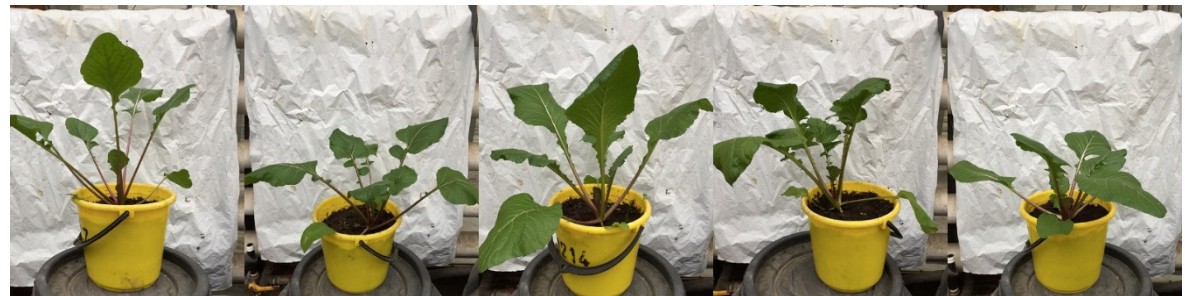

Substrate

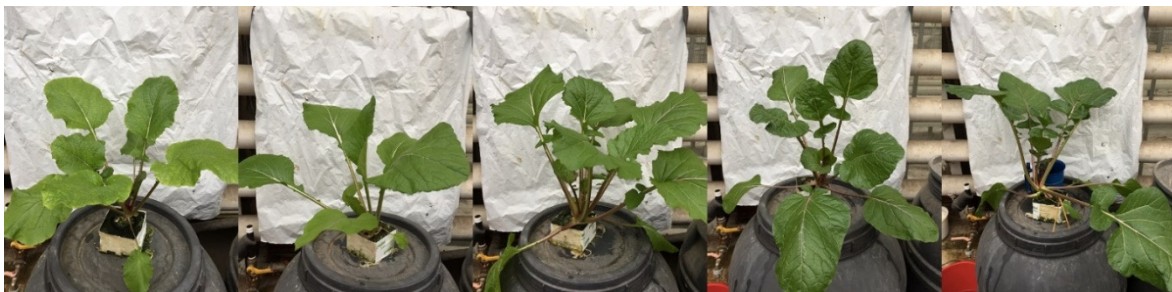

Hydroponic

**Figure 2.** Autumn view of plants (from left: N = 50, 70. 90, 110, and 130).

*2.1. Flowering Chinese Cabbage Harvest and Storage*

On the last day of every experiment (28th of May and 29th of October, respectively) the aboveground parts of plants were weighted. The parts of the fresh plant material from each combination were collected and freeze-dried at $-59\ °C$ (FreeZone, LANCONCO) then ground and stored in a freezer ($-18\ °C$) for the following analyses: colour measurements, chlorophyll and carotenoid content, total polyphenol content, total flavonoid content, and antioxidant activity.

### 2.2. Analysis of Macro- and Microelements in Plants

All analyses were conducted on the aerial parts of the plants. The samples were dried for 48 h at 45–50 °C to a steady mass and then ground. Before mineralization, the plant material was dried for 1 h at 105 °C. In order to determine the total content of N, P, K, Ca, Mg, and Na, the plant material (1 g) was digested in concentrated (96%, analytically pure) sulphuric acid (20 cm$^3$) with hydrogen peroxide (30%, analytically pure) [11]. For analyses of the total Fe, Mn, Zn, and Cu content, the plant material (2.5 g) was digested in a mixture of concentrated nitric (ultra-pure) and perchloric acids (analytically pure) at a 3:1 ratio (30 cm$^3$). After mineralization, the following measurements were taken: total N—the Kjeldahl distillation method in a Parnas Wagner apparatus; P—colorimetry with ammonia molybdate; and K, Ca, Mg, Na (results expressed in % DM—dried matter), Fe, Mn, Zn, and Cu (results expressed in mg·kg$^{-1}$ DM)—flame atomic absorption spectroscopy (FAAS) on a Carl Zeiss Jena 5 apparatus (Thornwood, NY, USA). The accuracy of the methods used for the chemical analyses and the precision of analytical measurements of nutrient levels was tested by analysing the LGC7162 reference material (LGC standards) with an average nutrient recovery of 96% (N, P, K, Ca, Mg, Fe, Mn, and Zn).

### 2.3. Colour Measurements

The colour of the flowering Chinese cabbage was measured on a CM-5 spectrophotometer (Konica Minolta) using the freeze-dried powdered sample with three replications. The samples were placed in the polystyrene containers; the layer of powder for the colour measurement was 1 cm. The conditions of the measurements were as follows: 10° angle of the observer and D65 as a source light. The colour was described in terms of L* (lightness), a* (red/green), and b* (yellow/blue), which are the colour space values according to standard CIE 1976 chromatic coordinates. The other determined chromatic attributes included chroma (C*) and hue angle (h*). Before each start-up, the spectrophotometer was calibrated automatically (internal white plate) and manually (black box for zero calibration).

### 2.4. Chlorophyll and Carotenoid Content Measurements

The total chlorophyll and carotenoid measurements were performed as previously described [12] based on the original methodology [13]. Briefly, the powdered flowering Chinese cabbage (0.125 g) was mixed with methanol (5 mL). After 5 min, the sample was filtrated and 80 μL of eluate was mixed with 3.92 mL of methanol and the absorbance scan was performed within the wavelength range of 410 nm to 700 nm on a Cary 1E UV–vis spectrophotometer (Varian, Belrose, Australia). The baseline was done for methanol. The contents of Chl a, Chl b, and carotenoids were calculated via an equation from [13] using the absorbance values at 665 nm, 652 nm, and 470 nm. All measurements were performed in triplicates.

### 2.5. TPC and TFC Measurements

The determination of the TPC was conducted by using a Cary 1E spectrophotometer. The procedure was a modified method described previously by Singleton and Rossi [14]. In brief, a freeze-dried sample of flowering Chinese cabbage was mixed with distilled water (0.125 g/5 mL). After 30 min, the sample was filtrated and 20 μL of the filtrate was mixed with 100 μL of Folin–Ciocalteu's reagent (FCR). The sample was incubated at room temperature in the dark; after 3 min, 300 μL of sodium carbonate solution (20% $w/v$) and distilled water up to the total volume of 2 mL were added. The sample was mixed again and left to incubate for 2 h in the dark at room temperature. Then, the absorbance was read at the wavelength of 765 nm against blank samples (performed as described above, but instead of the sample, the distilled water was added). The results were expressed as mg of gallic acid equivalents (GAE) per g of dried matter (DM).

The total flavonoid content (TFC) measurements were performed in the following manner: 100 μL of the sample (prepared as above) was mixed with 900 μL of $AlCl_3$ (2% in methanol) and left in the dark at room temperature. After 15 min of incubation, absorbance readings were taken at the wavelength of 410 nm. All measurements were performed in triplicates.

### 2.6. Antioxidant Activity Measurements

In order to evaluate the antioxidant activity of the flowering Chinese cabbage, three methods were applied in this experiment: the Trolox-equivalent antioxidant capacity (TEAC), 2,2-diphenyl-1-picrylhydrazyl (DPPH), and ferric-reducing antioxidant power (FRAP) assays, which showed various mechanisms of antioxidant action. The sample for the antioxidant activity measurements was prepared as follows: 0.25 g of the powdered freeze-dried flowering Chinese cabbage was extracted with 5 mL of 99% methanol for 30 min.

The TEAC assay is based on the antioxidant to scavenge the stable radial cation $ABTS^{•+}$ (2,2′-azinobis (3-ethylbenzothiazoline-6-sulfonic acid)) [15]. The $ABTS^{•+}$ radical cations were generated in potassium persulfate ($K_2S_2O_8$) for 12–16 h of incubation. Then, the $ABTS^{•+}$ solution was mixed with the PBS buffer (pH 7.4) to obtain the absorbance of the solution at 0.7. The PBS was prepared on the day of the analysis. Then, 10 μL of the sample was added to 990 μL of $ABTS^{•+}$ in PBS, mixed, and left to incubate for 6 min in the dark. The absorbance was read at 734 nm; the results were expressed as TEAC values in μmol per g of dried sample.

The DPPH assay was carried out according to the procedure in [16] with some modifications [12]. The sample was mixed with the DPPH in methanol (0.1 mmol) at a ratio of 1:100 (*v/v*). The reaction mixture was incubated in the dark at room temperature for 30 min, and the absorbance was read at 515 nm against the blank (pure methanol) using a Cary 1E spectrophotometer. The $DPPH^•$ radical-scavenging activity of the sample was expressed in μmol of Trolox equivalent (TE) per g of flowering Chinese cabbage.

The FRAP method is based on the chelating ability of the antioxidant; namely, the reduction of $Fe^{3+}$-tripyridyl-triazine (TPTZ) to a ferrous form by the antioxidant [17]. In brief, the sample was mixed with the TPTZ working solution at a ratio of 1:20 (*v/v*). The absorbance at 593 nm was read after 15 min of sample incubation in the dark. The results were expressed as mmol TE per g of the sample.

### 2.7. Statistical Analysis

The data were analysed using Statistica 13.3 software (StatSoft Inc., Tulsa, OK, USA). The results of the chemical analyses and plant-yield measurements were processed using ANOVA. The variance homogeneity was checked by a Cochran–Hartley–Bartlett test. The multiple comparison was based on a Duncan's test to determine the significant differences between samples. A principle component analysis (PCA) was performed to show the relationships between the variables and to gain more insight into the data. All analyses were done at a significance level of $\alpha = 0.05$.

## 3. Results

### 3.1. Plant Yields

Nitrogen nutrition positively impacted plant yields (Figure 3)—this effect was generally similar in both seasons but varied depending on the type of cultivation. For the mean of two seasons, the yield increased from N-50 to N-90. More intensive N nutrition (N-110 and N-130) did not significantly modify plant yields.

### 3.2. Macroelements

In the case of the substrate cultivation in both seasons, we found that there we a positive effect of the N nutrition level on the content of that nutrient in the aboveground parts of plants (Table 1).

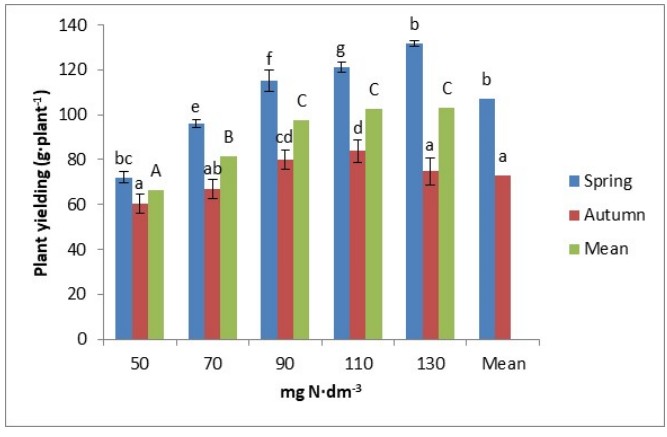

substrate

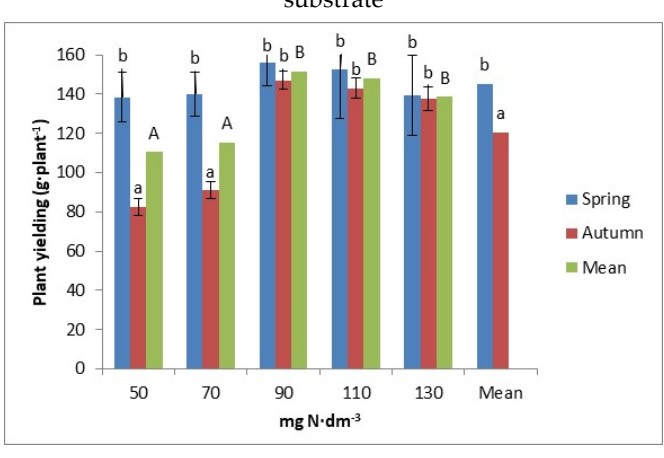

hydroponic

**Figure 3.** The yields of flowering Chinese cabbage grown in substrate (I) and hydroponically (II) with different N levels (g·plant$^{-1}$). The values denoted with the same lowercase letter did not significant differ at $p = 0.05$. The mean values denoted with the same capital letter did not significant differ at $p = 0.05$.

In hydroponic conditions, the N content was generally stable and independent of the intensity of the N nutrition and the period of cultivation. In autumn cultivation in the substrate, the N content was about 60% higher compared to that in the spring. However, there were no statistical differences in N leaf content among the combinations in hydroponic cultivation, but generally that type of cultivation promoted higher N levels compared with substrate cultivation. For P, the mean values for season (in substrate cultivation) were similar (spring vs. autumn); meanwhile, in the case of hydroponics, the highest mean content (+about 50%) was determined for autumn cultivation. The highest K status was determined for autumn cultivation independent of the type of cultivation. Similar to the case of K, the time of cultivation varied the status of Ca (the highest content was determined for autumn). In the case of the substrate content of Ca in plants, it was less stable compared to hydroponics. In the autumn period, the content of Mg was lower compared to that in the spring period. Similar to the case of Ca, the Mg contents in plants were stable in the case of hydroponics compared to cultivation in the substrate (the differences were in the spring).

### 3.3. Microelements and Sodium

Generally, increasing the intensity of the N nutrition improved the nutrient status with microelements (Table 2). The content of Fe was higher in the autumn period (for both types of cultivation); meanwhile, in the case of Mn, the highest contents were determined for the spring (hydroponics) or autumn period (substrate). The opposite situation was found for Cu content. The content of Na, independent of the cultivation type, was higher in autumn compared to spring.

**Table 1.** The effect of nitrogen level and planting date (season) on the content of macroelements (% DM) in aboveground parts of plants grown in substrate and hydroponically.

| N Level | N | | | P | | | K | | | Ca | | | Mg | | |
|---|---|---|---|---|---|---|---|---|---|---|---|---|---|---|---|
| | Spring * | Autumn * | Mean ** | Spring | Autumn | Mean | Spring | Autumn | Mean | Spring | Autumn | Mean | Spring | Autumn | Mean |
| Substrate | | | | | | | | | | | | | | | |
| N-50 | 1.92 [a] | 4.15 [d] | 3.03 [a] | 0.46 [a] | 0.59 [c] | 0.53 [b] | 4.62 [a] | 7.70 [d] | 6.16 [a] | 2.13 [a] | 2.40 [ab] | 2.27 [ab] | 2.13 [bc] | 0.54 [a] | 1.33 [ab] |
| N-70 | 2.14 [a] | 4.11 [d] | 3.12 [a] | 0.50 [ab] | 0.47 [ab] | 0.48 [ab] | 4.96 [ab] | 7.40 [cd] | 6.17 [a] | 1.92 [a] | 2.28 [ab] | 2.10 [a] | 1.92 [b] | 0.49 [a] | 1.21 [a] |
| N-90 | 2.84 [b] | 4.32 [de] | 3.58 [b] | 0.46 [a] | 0.47 [ab] | 0.47 [a] | 4.62 [a] | 6.93 [c] | 5.78 [a] | 2.01 [a] | 2.64 [b] | 2.32 [ab] | 2.01 [b] | 0.52 [a] | 1.26 [ab] |
| N-110 | 3.19 [bc] | 4.43 [de] | 3.81 [bc] | 0.50 [ab] | 0.44 [a] | 0.47 [a] | 5.01 [ab] | 7.05 [cd] | 6.03 [a] | 2.39 [ab] | 3.28 [c] | 2.83 [c] | 2.39 [c] | 0.49 [a] | 1.44 [b] |
| N-130 | 3.50 [c] | 4.71 [e] | 4.10 [c] | 0.54 [bc] | 0.47 [ab] | 0.51 [ab] | 5.38 [b] | 7.17 [cd] | 6.28 [a] | 2.40 [ab] | 2.66 [b] | 2.53 [bc] | 2.40 [c] | 0.51 [a] | 1.45 [b] |
| Mean ** | 2.71 [a] | 4.34 [b] | | 0.49 [a] | 0.49 [a] | | 4.92 [a] | 7.25 [b] | | 2.17 [a] | 2.65 [b] | | 2.17 [b] | 0.51 [a] | |
| ±SD | 0.67 | 0.38 | | 0.05 | 0.07 | | 0.47 | 0.48 | | 0.32 | 0.44 | | 0.32 | 0.04 | |
| Hydroponic | | | | | | | | | | | | | | | |
| N-50 | 4.39 [ab] | 4.53 [ab] | 4.46 [a] | 0.50 [a] | 0.59 [cd] | 0.55 [a] | 4.96 [a] | 8.21 [c] | 6.59 [a] | 2.01 [a] | 2.55 [ab] | 2.28 [a] | 2.01 [b] | 1.14 [a] | 1.57 [a] |
| N-70 | 4.34 [ab] | 4.67 [ab] | 4.50 [a] | 0.52 [ab] | 0.64 [cd] | 0.58 [a] | 5.20 [a] | 8.03 [c] | 6.62 [a] | 2.19 [ab] | 2.49 [ab] | 2.34 [a] | 2.19 [b] | 1.10 [a] | 1.64 [a] |
| N-90 | 4.48 [ab] | 4.36 [ab] | 4.42 [a] | 0.52 [a] | 0.58 [bc] | 0.55 [a] | 5.17 [a] | 8.06 [c] | 6.62 [a] | 2.02 [a] | 2.53 [ab] | 2.28 [a] | 2.02 [b] | 1.11 [a] | 1.56 [a] |
| N-110 | 4.18 [a] | 4.53 [ab] | 4.35 [a] | 0.52 [ab] | 0.61 [cd] | 0.57 [a] | 5.23 [a] | 7.06 [b] | 6.14 [a] | 2.01 [a] | 2.68 [b] | 2.35 [a] | 2.01 [b] | 1.05 [a] | 1.53 [a] |
| N-130 | 4.62 [ab] | 4.95 [b] | 4.78 [a] | 0.53 [ab] | 0.64 [d] | 0.59 [a] | 5.26 [a] | 7.36 [bc] | 6.31 [a] | 2.20 [ab] | 2.49 [ab] | 2.35 [a] | 2.20 [b] | 1.05 [a] | 1.63 [a] |
| Mean | 4.40 [a] | 4.61 [a] | | 0.52 [a] | 0.61 [b] | | 5.16 [a] | 7.74 [b] | | 2.09 [a] | 2.55 [b] | | 2.09 [b] | 1.09 [a] | |
| ±SD | 0.43 | 0.22 | | 0.03 | 0.04 | | 0.38 | 0.72 | | 0.26 | 0.24 | | 0.26 | 0.07 | |

For all the tables: * Values with the same letter within a column and type of cultivation were not significantly different ($p > 0.05$, Duncan's test). ** Mean value for all the treatments in a column with the same letter within the column and raw for the parameter were not significantly different ($p > 0.05$, Duncan's test). Details on the standard deviations (SD) are shown in the Supplementary Materials (Tables S1–S5).

**Table 2.** The effect of nitrogen level and planting date (season) on the content of microelements (mg·kg$^{-1}$ DM) and sodium (mg·kg$^{-1}$ DM) in aboveground parts of plants grown in substrate and hydroponically.

| N Level | Fe | | | Mn | | | Zn | | | Cu | | | Na | | |
|---|---|---|---|---|---|---|---|---|---|---|---|---|---|---|---|
| | Spring * | Autumn * | Mean ** | Spring | Autumn | Mean | Spring | Autumn | Mean | Spring | Autumn | Mean | Spring | Autumn | Mean |
| | | | | | | Substrate | | | | | | | | | |
| N-50 | 126.1 ab | 111.4 a | 118.8 a | 28.1 a | 39.0 bcd | 33.6 a | 42.8 a | 47.9 abc | 45.4 a | 7.55 cd | 5.20 a | 6.37 a | 0.09 a | 0.19 b | 0.14 a |
| N-70 | 117.4 ab | 154.1 b | 135.8 a | 32.8 ab | 38.4 bcd | 35.6 ab | 47.3 abc | 46.4 a | 46.8 a | 8.41 de | 5.07 a | 6.74 ab | 0.11 a | 0.22 b | 0.16 ab |
| N-90 | 126.1 ab | 119.5 ab | 122.8 a | 38.8 bcd | 40.6 bcd | 39.7 bc | 59.5 d | 46.7 ab | 53.1 b | 7.03 bcd | 6.07 abc | 6.55 ab | 0.12 a | 0.28 c | 0.20 b |
| N-110 | 115.3 ab | 147.6 ab | 131.5 a | 36.4 abc | 43.8 cd | 40.1 bc | 55.6 cd | 51.0 abcd | 53.3 b | 9.24 e | 5.77 ab | 7.51 b | 0.18 b | 0.30 c | 0.24 c |
| N-130 | 152.5 b | 209.4 c | 181.0 b | 40.3 bcd | 46.7 d | 43.5 c | 57.2 d | 55.3 bcd | 56.2 b | 7.84 de | 6.31 abc | 7.07 ab | 0.20 b | 0.31 c | 0.25 c |
| Mean ** | 127.5 a | 148.4 b | | 35.3 a | 41.7 b | | 52.5 a | 49.5 a | | 8.01 b | 5.68 a | | 0.14 a | 0.26 b | |
| ±SD | 22.6 | 41.6 | | 7.81 | 5.87 | | 6.22 | 6.14 | | 1.51 | 0.83 | | 0.05 | 0.06 | |
| | | | | | | Hydroponic | | | | | | | | | |
| N-50 | 86.8 a | 108.4 bc | 97.6 a | 73.95 e | 44.45 a | 59.20 ab | 39.00 c | 31.67 a | 35.33 a | 4.03 ab | 5.08 cd | 4.56 a | 0.18 a | 0.25 d | 0.22 a |
| N-70 | 92.1 ab | 106.4 abc | 99.3 a | 65.08 cde | 51.75 ab | 58.42 a | 32.29 ab | 45.17 d | 38.73 a | 3.92 ab | 5.36 de | 4.64 a | 0.19 a | 0.25 d | 0.22 a |
| N-90 | 99.6 abc | 118.2 c | 108.9 a | 66.29 cde | 57.63 bc | 61.96 abc | 37.02 bc | 47.78 d | 42.40 b | 3.78 a | 5.89 e | 4.84 a | 0.19 ab | 0.25 d | 0.22 a |
| N-110 | 93.2 ab | 117.1 c | 105.1 a | 66.62 cde | 63.28 cd | 64.95 bc | 30.72 a | 59.97 e | 45.35 bc | 3.71 a | 5.97 e | 4.84 a | '0.19 a | 0.23 bcd | 0.21 a |
| N-130 | 107.8 bc | 113.9 c | 110.8 a | 70.99 de | 60.28 c | 65.63 c | 32.15 ab | 59.83 e | 45.99 c | 4.57 bc | 6.65 f | 5.61 b | 0.20 abc | 0.24 cd | 0.22 a |
| Mean | 95.9 a | 112.8 b | | 68.59 b | 55.48 a | | 34.23 a | 48.89 b | | 4.00 a | 5.79 b | | 0.19 a | 0.24 b | |
| ±SD | 12.07 | 16.38 | | 5.51 | 9.49 | | 4.35 | 9.90 | | 0.44 | 0.62 | | 0.02 | 0.02 | |

### 3.4. Colour Parameters

The results of all analysed parameters are shown in Table 3. The effects of N nutrition and season (planting date) were statistically significant for the colour parameters (Table 3). The highest lightness was determined for the spring samples within the substrate as well as in hydroponic cultivation. The autumn samples from hydroponic conditions showed higher L* values (48.13) than the autumn samples from the substrate type of cultivation (46.92). The a* and b* colour-coordinate values were in a similar range in the substrate and in hydroponic conditions, showing values lower for autumn when compared to the spring samples. However, when analysing the effect of N nutrition, the pattern of colour changes varied. In our experiment, in both the spring and autumn seasons in the substrate experiment, the greenness of the Chinese cabbage increased with an increase in the nitrogen level (from N-50 to N-110 in spring and to N-90 in autumn). With a further increase in the N level, the greenness decreased (higher a* values).

**Table 3.** The effect of nitrogen level and planting date (season) on the colour parameters (L*—lightness, a*, b*—colour chromatic coordinates, h*—hue angle, C*—chroma) of the aboveground parts of plants grown in substrate and hydroponically.

| N Dose | L* (D65) | | a* (D65) Green | | b* (D65) Yellow | | h* (°) (D65) | | C* | |
|---|---|---|---|---|---|---|---|---|---|---|
| | Spring | Autumn | Spring | Autumn | Spring | Autumn | Spring | Autumn | Spring | Autumn |
| | | | | | Substrate | | | | | |
| N-50 | 47.50 [g] | 43.82 [b] | −8.66 [c] | −7.32 [j] | 22.36 [j] | 20.21 [f] | 111.16 [c] | 109.91 [a] | 23.98 [j] | 21.50 [f] |
| N-70 | 45.16 [c] | 48.27 [h] | −7.99 [g] | −7.40 [i] | 20.43 [g] | 19.14 [c] | 111.36 [d] | 111.14 [b] | 21.94 [g] | 20.52 [c] |
| N-90 | 45.5 [e] | 48.32 [i] | −9.17 [b] | −8.60 [d] | 21.32 [h] | 18.47 [a] | 113.29 [h] | 114.98 [j] | 23.21 [h] | 20.37 [a] |
| N-110 | 45.63 [f] | 48.97 [j] | −9.19 [a] | −8.31 [e] | 21.38 [i] | 19.20 [d] | 113.26 [g] | 113.41 [i] | 23.27 [i] | 20.93 [d] |
| N-130 | 43.65 [a] | 45.24 [d] | −8.03 [f] | −7.81 [h] | 19.80 [e] | 18.89 [b] | 112.08 [e] | 112.47 [f] | 21.36 [e] | 20.44 [b] |
| | | | | | Hydroponic | | | | | |
| N-50 | 47.42 [f] | 49.01 [j] | −8.36 [d] | −7.64 [g] | 19.47 [e] | 18.46 [b] | 113.23 [gh] | 112.48 [d] | 21.18 [e] | 19.98 [b] |
| N-70 | 44.75 [c] | 48.82 [i] | −8.35 [d] | −8.18 [e] | 19.47 [e] | 21.11 [f] | 113.20 [g] | 111.18 [a] | 21.18 [e] | 22.64 [f] |
| N-90 | 43.84 [a] | 45.68 [e] | −8.34 [d] | −8.97 [c] | 19.40 [d] | 22.28 [g] | 113.26 [h] | 111.91 [b] | 21.12 [d] | 24.02 [g] |
| N-110 | 45.28 [d] | 48.66 [h] | −9.49 [a] | −7.54 [h] | 22.67 [i] | 16.73 [a] | 112.71 [e] | 114.24 [i] | 24.57 [i] | 18.35 [a] |
| N-130 | 44.35 [b] | 48.46 [g] | −9.25 [b] | −8.08 [f] | 22.52 [h] | 19.13 [c] | 112.32 [c] | 112.91 [f] | 24.35 [h] | 20.77 [c] |

### 3.5. Chorophyll and Carotenoid Content

Table 4 includes data on the chlorophyll and carotenoid content. When analysing the chlorophyll content for the substrate cultivation, a significant effect of the N fertilisation was observed for the Chl a, sum of chlorophylls, and Chl a/Chl b parameters. A significant effect of the season (planting date) was noted for almost all indices in Table 4 apart from the chlorophyll ratio (Chl a/Chl b). The highest values for chlorophylls and carotenoids were generally at the levels of 90–130 mg·dm$^{-3}$ of N.

For hydroponic cultivation, only the planting date was statistically significant except for its effect on the Chl a/Chl b index. The highest content of chlorophylls was obtained in the hydroponic cultivation for the spring samples. The contents of chlorophylls and carotenoids in the autumn samples were similar for the substrate and hydroponic types of cultivation.

**Table 4.** The effect of nitrogen level and planting date (season) on the chlorophyll (Chl a and Chl b) and carotenoid (x + c) content in aboveground parts of plants grown in substrate and hydroponically. DM—dried matter.

| N Level | Chl a (mg/g DM) | | Chl b (mg/g DM) | | Carotenoid (x + c) (mg/g DM) | | Chl a + Chl b (mg/g DM) | | Chl a/Chl b | | (Chl a + Chl b)/(x + c) | |
|---|---|---|---|---|---|---|---|---|---|---|---|---|
| | Spring | Autumn | Spring | Autumn | Spring | Autumn | Spring | Autumn | Spring | Autumn | Spring | Autumn |
| Substrate | | | | | | | | | | | | |
| N-50 | 5.14 [a] | 5.50 [a] | 1.43 [a] | 1.44 [a] | 1.40 [b] | 1.32 [ab] | 6.57 [a] | 6.94 [a] | 3.77 [a] | 3.99 [a] | 4.65 [a] | 5.28 [ab] |
| N-70 | 5.47 [a] | 4.64 [a] | 1.59 [ab] | 1.52 [ab] | 1.34 [ab] | 0.98 [a] | 7.06 [a] | 6.16 [a] | 3.44 [a] | 3.05 [a] | 5.29 [ab] | 6.31 [b] |
| N-90 | 8.13 [b] | 5.26 [a] | 2.16 [b] | 1.64 [ab] | 2.01 [c] | 1.09 [ab] | 10.29 [b] | 6.90 [a] | 3.76 [a] | 3.21 [a] | 5.19 [ab] | 6.34 [b] |
| N-110 | 7.62 [b] | 5.05 [a] | 2.00 [bc] | 1.48 [ab] | 2.03 [c] | 1.07 [ab] | 9.62 [b] | 6.53 [a] | 3.81 [a] | 3.49 [a] | 4.74 [a] | 6.18 [b] |
| N-130 | 8.33 [b] | 4.88 [a] | 2.29 [c] | 1.32 [a] | 1.95 [c] | 1.06 [ab] | 10.62 [b] | 6.20 [a] | 3.64 [a] | 3.79 [a] | 5.47 [ab] | 5.95 [b] |
| Hydroponic | | | | | | | | | | | | |
| N-50 | 7.46 [b] | 4.68 [a] | 1.98 [b] | 1.38 [a] | 1.71 [b] | 1.00 [a] | 9.44 [b] | 6.06 [a] | 3.77 [a] | 3.41 [a] | 5.55 [abc] | 6.06 [abc] |
| N-70 | 7.72 [cb] | 5.03 [a] | 2.10 [b] | 1.27 [a] | 1.78 [bc] | 1.21 [a] | 9.82 [bc] | 6.31 [a] | 3.68 [a] | 4.09 [a] | 5.53 [abc] | 5.24 [a] |
| N-90 | 8.75 [c] | 4.87 [a] | 2.36 [b] | 1.42 [a] | 1.99 [bc] | 1.02 [a] | 11.11 [c] | 6.29 [a] | 3.70 [a] | 3.46 [a] | 5.61 [abc] | 6.14 [bc] |
| N-110 | 8.31 [cb] | 4.84 [a] | 2.28 [b] | 1.36 [a] | 1.87 [bc] | 0.98 [a] | 10.59 [bc] | 6.20 [a] | 3.68 [a] | 3.71 [a] | 5.67 [abc] | 6.30 [c] |
| N-130 | 8.90 [c] | 5.44 [a] | 2.42 [b] | 1.32 [a] | 2.02 [c] | 1.26 [a] | 11.31 [c] | 6.77 [a] | 3.68 [a] | 4.24 [a] | 5.61 [abc] | 5.42 [ab] |

### 3.6. Phenolic Content and the Antioxidant Activity

The effects of the planting date and nitrogen level on the antioxidant activity and phenolic content are shown in Table 5. These effects were statistically significant in the hydroponic cultivation, whereas in the substrate, both effects were significant for the DPPH, FRAP, and TPC values. In the substrate, the TEAC values resulted in significant differences depending on the N nutrition, while the TFC values were differentiated only by season. In general, the spring samples were characterised by higher values for the antioxidant activity and phenolic content than the autumn samples; the highest values were observed for the substrate type of cultivation. Moreover, various N levels were required to obtain the maximum values for those bioactive compounds and the antioxidant activity parameters. Generally, in the substrate system, the antioxidant activity of the samples was the highest at a medium level of nitrogen fertilisation (N-90 and/or N-110) as an average over season. Additionally, the total phenolic content (TPC) of the samples was the highest at a medium nitrogen concentration (N-90), whereas the flavonoid content was the highest for N-70 and N-90.

**Table 5.** The effect of nitrogen level and planting date (season) on the antioxidant activity (TEAC, DPPH, and FRAP) and phenolic content (TPC and TFC) of the aboveground parts of plants grown in substrate and hydroponically. TEAC—Trolox equivalent antioxidant capacity, DPPH—diphenyl-picryl hydrazyl, FRAP—ferric-reducing antioxidant power, TPC—total polyphenol content, TFC—total flavonoid content, TE—Trolox equivalent, QE—quercetin equivalent.

| N Dose | TEAC (µmol/g) | | DPPH (µmol TE/g) | | FRAP (mmol TE/g) | | TPC (mg GAE/g) | | TFC (mg QE/g) | |
|---|---|---|---|---|---|---|---|---|---|---|
| | Spring | Autumn | Spring | Autumn | Spring | Autumn | Spring | Autumn | Spring | Autumn |
| Substrate | | | | | | | | | | |
| N-50 | 396.54 [a] | 392.85 [a] | 120.19 [bc] | 108.78 [b] | 33.23 [d] | 24.59 [b] | 6.28 [c] | 4.22 [a] | 1.89 [d] | 1.45 [bc] |
| N-70 | 417.17 [b] | 414.83 [b] | 148.00 [d] | 107.41 [b] | 32.83 [d] | 24.42 [b] | 6.44 [c] | 4.21 [a] | 3.90 [g] | 1.16 [ab] |
| N-90 | 501.87 [ef] | 465.38 [d] | 188.21 [f] | 125.56 [c] | 44.75 [e] | 27.58 [c] | 8.96 [e] | 6.37 [c] | 2.73 [ef] | 1.47 [c] |
| N-110 | 484.75 [e] | 511.69 [fg] | 199.31 [f] | 89.41 [a] | 46.11 [e] | 21.26 [a] | 7.37 [d] | 3.61 [a] | 2.86 [f] | 1.13 [a] |
| N-130 | 437.90 [c] | 524.21 [g] | 164.73 [e] | 106.98 [b] | 45.45 [e] | 23.64 [b] | 8.81 [e] | 5.16 [b] | 2.48 [e] | 1.29 [abc] |
| Hydroponic | | | | | | | | | | |
| N-50 | 410.87 [ab] | 447.63 [bcde] | 101.48 [a] | 121.05 [c] | 37.16 [de] | 21.56 [a] | 6.01 [cde] | 5.31 [abcd] | 1.40 [bcd] | 0.97 [a] |
| N-70 | 363.92 [a] | 419.34 [bc] | 148.11 [d] | 103.95 [a] | 34.75 [cd] | 28.28 [abc] | 6.32 [de] | 4.88 [ab] | 1.62 [de] | 1.25 [abc] |
| N-90 | 434.20 [bcd] | 481.87 [de] | 140.79 [d] | 111.37 [abc] | 38.04 [de] | 30.12 [bc] | 8.55 [f] | 5.53 [bcd] | 1.81 [e] | 1.11 [ab] |
| N-110 | 467.72 [cde] | 497.63 [e] | 151.29 [d] | 119.44 [bc] | 39.31 [de] | 25.52 [ab] | 8.18 [f] | 6.73 [e] | 1.33 [bc] | 1.31 [bc] |
| N-130 | 480.25 [de] | 423.21 [bc] | 170.05 [e] | 105.87 [ab] | 43.03 [e] | 25.97 [ab] | 4.95 [abc] | 4.38 [a] | 1.01 [a] | 1.41 [cd] |

In the hydroponic system, the highest values for the antioxidant activity of the samples cultivated in spring were obtained at high N levels (N-110 and/or N-130). For the autumn samples, the highest antioxidant activity was noted at various N levels (the low and medium N nutrition levels N-50, N-90, and N-110). The phenolic content was the highest for N-90 in the spring and N-110 in the autumn samples, whereas the flavonoid content was the highest at various N levels.

*3.7. Principle Component Analysis (PCA)*

The PCA was performed in order to show the structure of the relationships between all variables and to show the distribution of the scores (samples) according to these variables. Before the analysis, the data matrix was standardised and the r-Pearson correlation coefficients (Tables S6 and S7 in the Supplementary Materials) were calculated.

Based on the eigenvalue plot, the first five principal components (PCs) were derived with the eigenvalue greater than 1, which together explained 84.2% of total variance in the substrate and 83% in the hydroponic datasets. Figures 4 and 5 show the projection of the variables (a) and scores (b) onto the factor plane with the first principle component (PC1) and the second principle component (PC2) on the X and Y axes for both the substrate (Figure 4) and hydroponic (Figure 5) experiments, respectively.

The loaded values, which showed correlations between the components and variables, indicated that in the substrate, the highest positive correlations were observed between PC1 and N (0.74), K (0.88), and Na (0.72), all of which were located on the right side of the X axis (Figure 4a). The variables negatively correlated with the PC1 (left side of the X axis) were as follows: Mg (−0.93), Cu (−0.70), DPPH (−0.88), FRAP (−0.93), TPC (−0.84), TFC (−0.80), b* (−0.78), C (−0.82), Chl a (−0.78), carotenoids (−0.89), and sum of chlorophylls (−0.76). The distribution of the samples is shown in Figure 4b. As evidenced by the analysis of Figure 4b, the samples could be divided into three main clusters. Along with the PC1 axis, the first cluster of the autumn samples was placed on the positive side of the axis, which indicated the samples with a high value of N, K, and Na. Those samples were placed opposite to the other two clusters showing spring samples (negative PC1 values), which both were characterised by the highest values of Mg and Cu and the highest DPPH and FRAP antioxidant activity, TPC and TFC values, as well as chlorophylls and carotenoid contents. Thus, based on PC1, the dataset could be divided in terms of season. PC2 was mainly correlated with the TEAC antioxidant activity (−0.77) and h colour parameter (−0.73); it divided the spring samples well into two separate groups. The second cluster consisted of the spring samples subjected to a high intensity of N nutrition (from 90 to 130 mg·dm$^3$). That cluster was placed on the negative Y-axis; the samples were characterised by the high TEAC and h values. The third cluster, namely the spring samples subjected to N-50 and N-70 levels, was located on the positive side of the PC2 axis (low levels of the TEAC and h parameters) and was described by the negative PC1 and positive PC2 values. Although all of the autumn samples were distributed within the first cluster, they could also be divided according the TEAC and h values (PC2), similar to the spring samples. The low levels of N nutrition (50 and 70 mg·dm$^3$) caused low TEAC and h values of the samples, and the opposite high levels of N (90–130 mg·dm$^3$) caused high TEAC and h values.

Next, in the hydroponic experiment we observed that PC1 was highly, positively correlated (right side of the axis in Figure 5a) with the P (0.83), K (0.89), Ca (0.72), Na (0.72), Cu (0.84), and L* colour parameter (0.84) and was negatively correlated (left side of the X-axis in Figure 5a) with the Mg (−0.89), DPPH (−0.72), FRAP (−0.89), chlorophylls (from −0.92 to −0.96), and carotenoids (−0.94). The samples were divided into three distinct clusters. Along PC1, they were distributed in terms of season (Figure 5b).

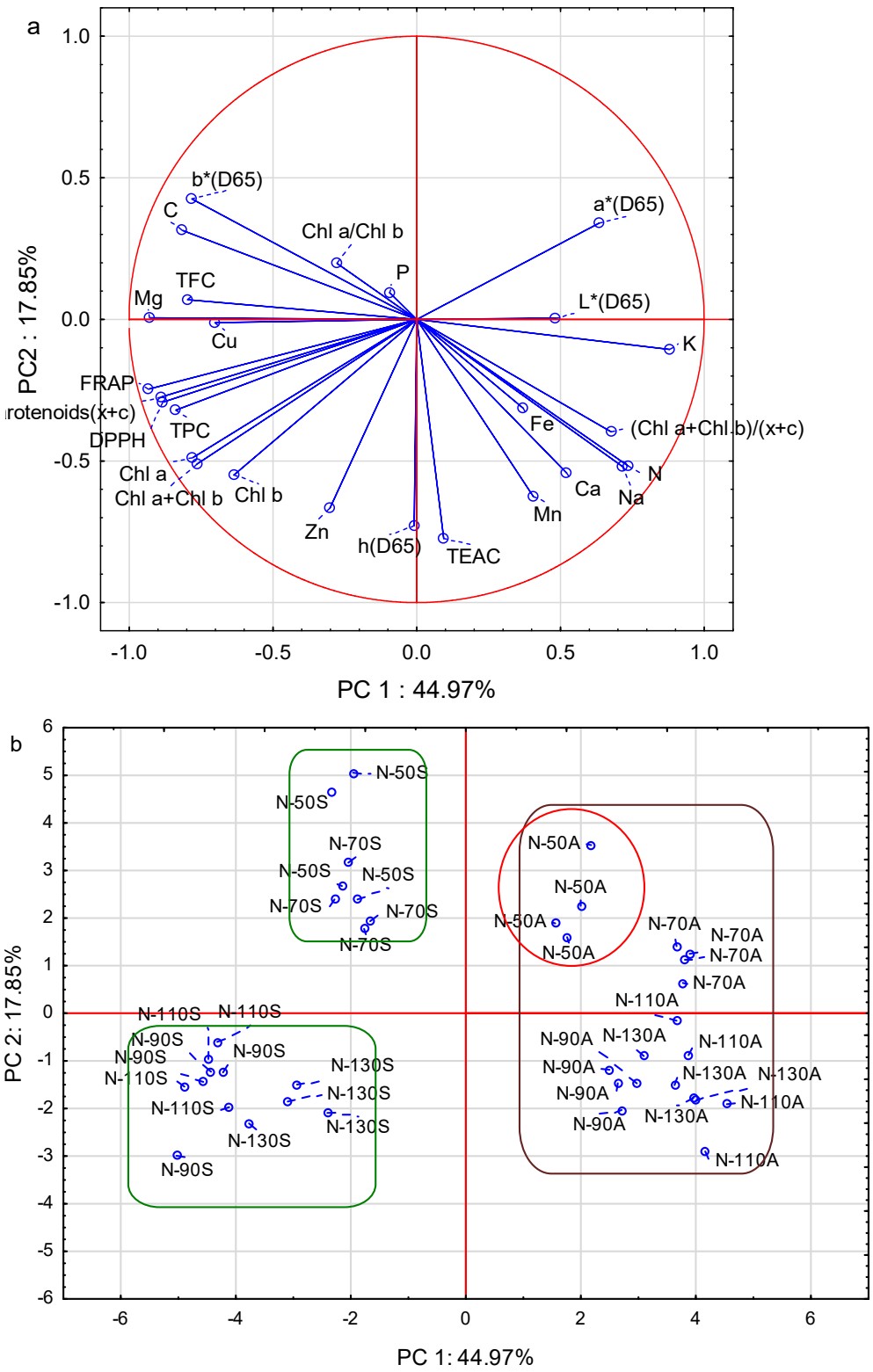

**Figure 4.** Projections of the (**a**) variables and (**b**) scores onto the factor plane defined by principal components (PC1 and PC2) for substrate experiment. N-50–N-130—nitrogen nutrition (from 50 to 130 mg·dm$^{-3}$); S—spring; A—autumn.

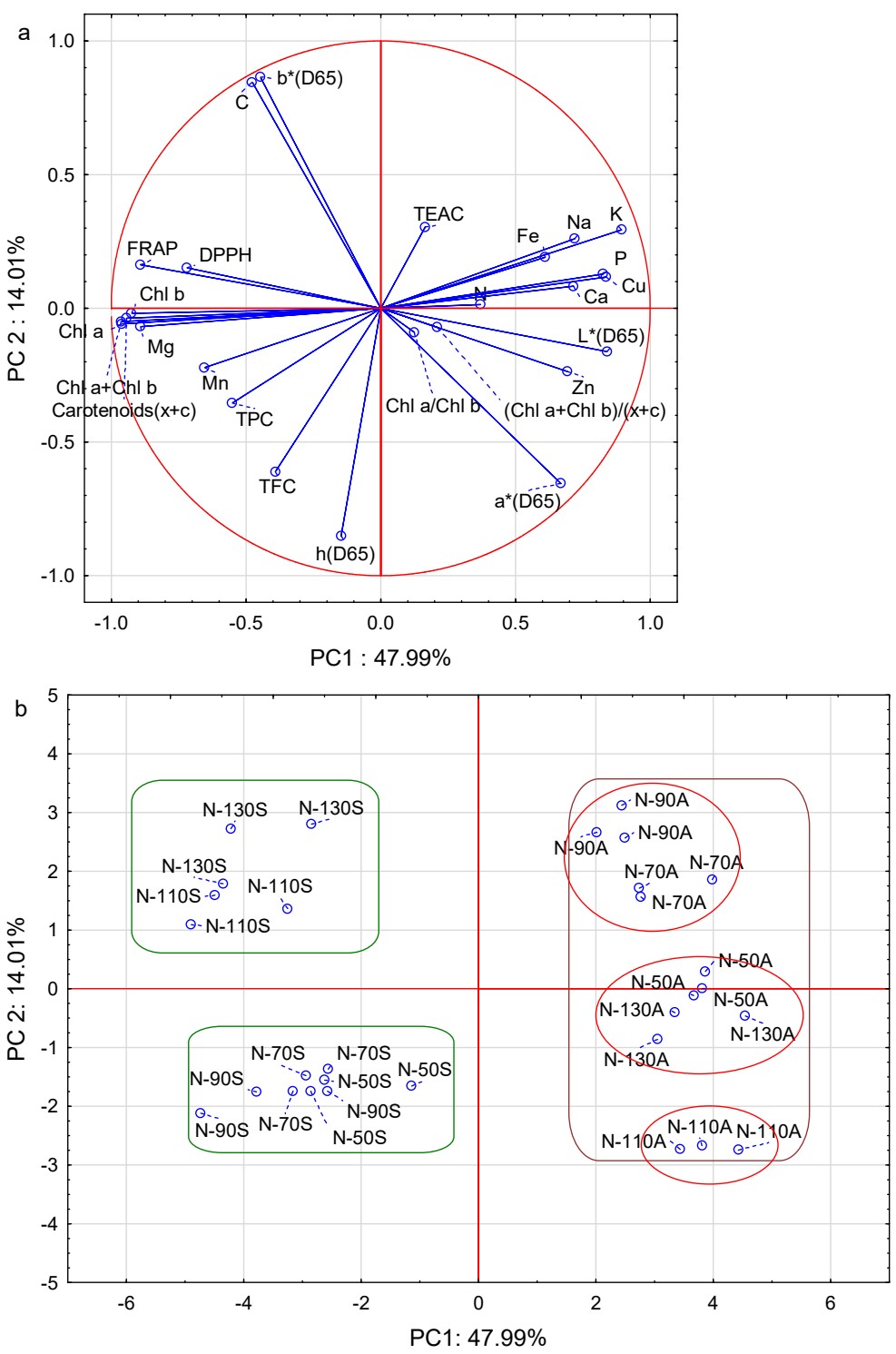

**Figure 5.** Projections of the (**a**) variables and (**b**) scores onto the factor plane defined by principal components (PC1 and PC2) for hydroponic experiment. N-50–N-130—nitrogen nutrition (from 50 to 130 mg·dm$^{-3}$); S—spring; A—autumn.

　　　　The autumn samples were placed on the right side of the axis with positive PC1 values, and thus had high contents of P, K, Ca, Na, Cu, high L* values, a low content of Mg, as well as low DPPH and FRAP antioxidant activity. Those samples constituted the first cluster. The spring samples were located on the opposite side of the PC1 axis. Moreover, since the b*, C, and h colour coordinates predominated PC2 (loaded values of 0.86, 0.85, and -0.85, respectively), the spring samples were divided into two clusters. The spring samples

subjected to a low intensity of N nutrition (from 50 to 90 mg·dm$^{-3}$) were placed on the negative side of the Y-axis and showed high h values (second cluster), whereas the spring samples treated with a high intensity of N nutrition (110 and 130 mg·dm$^{-3}$) with positive PC2 values showed high b* and C values (third cluster). However, no distinct clusters could be distinguished within the autumn samples (as was observed for the spring samples); they varied along with the PC2 values in terms of the b*, h, and C colour coordinates and could be classified into three subgroups (Figure 5b) as follows: N-110A low b* and C values and high h value; N-130A, N-50A moderate b*, C, and h values; and N-70A, N-90A high b* and C values and low h value.

## 4. Discussion

### 4.1. Plant Yields, Chlorophyll Content, and Colour

Chlorophyll contains N, which makes this element an important factor in the development of the photosynthetic apparatus [18]. The N supply and the ability of plants to respond to N have an influence on photosynthesis. Overall, additional N usually promotes plant growth, but the nature of this response depends on the patterns of plant nitrogen allocation [19]. Our studies showed a significant yield-generating effect of that nutrient. The increase in the yield was probably due to the improvement in the functioning of biochemical processes, including photosynthesis. Zhao et al. [20] reported on a closely positive relationship between increasing N content in leaves, leaf chlorophyll concentration, and photosynthesis. The strong positive correlations were also confirmed by many researchers [20–22].

In relation to environmental pollution, it is important to improve the efficiency of N nutrition. In our studies, the nutrient had a strong effect on the yields and some physicochemical properties of flowering Chinese cabbage—the optimal nitrogen level was 90 mg·dm$^{-3}$ (for mean values), but more intensive N-nutrition did not improve plant yields. In a previous study, Xie et al. [10] pointed out that the optimal nitrogen dose was 440–490 kg ha$^{-1}$ (mod. 220–245 mg·dm$^{-3}$ in 20 cm layer), which was a balance between the economic benefit and quality of flowering Chinese cabbage. However, it was determined in the case of field conditions with potential losses of N (leaching in the deeper layer). In our studies, in the autumn season, a higher intensity of N nutrition (>90 mg dm$^{-3}$) had a reducing effect on the yields of plants, which indicated that the nutritional requirements were lower than the fertilization requirements. The effective level of the use of nitrogen in soil is usually approximately 40–50%. Our findings were consistent with the results of other studies. The nitrogen fertilizer increased the yield of flowering Chinese cabbage by 41.80%, but excessive doses could reduce the yield [23]. Our findings were also consistent with the results of a study on Chinese cabbage conducted by Krężel and Kołota [24], who observed higher head yields when the nitrogen doses were increased from 50 to 150 kg N ha$^{-1}$ but a decreased yield at doses of 200 kg N ha$^{-1}$ (mod. 100 mg·dm$^{-3}$ in 20 cm layer of soil). From a practical point of view, excessive nitrogen nutrition does not improve yield; instead, it might cause environmental problems such as soil acidification and biological degradation [25].

Wang et al. [26] proved that the combination of organic fertiliser and biochar promoted the highest vegetative sprout lengths and increased the yield of flowering Chinese cabbage in Northwest China by 43.5%. In terms of N utilization, the key point is the content of nitrate levels in fertiliser. The partial replacement of N-NO$_3$ (20%) with the reduced form of N (NH$_4$) increased the fresh weight of Chinese kale depending on cultivars by 8.0–18.1% (respectively for 'Lvbao' and 'Zaobao') [27]. However, a high percentage of ammonium decreased the biomass of flowering Chinese cabbage; the experiment that applied 25%ammonium showed the best balance on the plants' biomass and nutrient uptake [28]. Zhu et al. [29] reported a similar result that appropriate N-NH$_4$/N-NO$_3$ ratios may improve N absorption and assimilation and thus promote the growth of flowering Chinese cabbage. They inferred that the biochemical reason that caused plant-growth inhibition via addition of excessive ammonium may have been a reduction in rhizosphere

acidification and ammonia toxicity. However, the ratio of ammonium and nitrate may vary according to different plants; for example, for blueberry seedling growth, the optimal ratio can be 5/1. Thus, further studies may focus on exploring the optimal nitrogen utilisation concerning forms of N fertilisers and co-present ions.

N is required in the highest quantities during stages of plant growth, which markedly affects the Rubisco content and photosynthesis [30,31]. The photosynthesis of leaves is related to N because the proteins of the Calvin cycle and thylakoids provide the majority of leaf nitrogen [31]. The ratio between the net to photosynthesis and leaf N content defines the photosynthetic N use efficiency as the amount of $CO_2$ fixed per unit of leaf N, which is varied as a function of the leaf N content [32,33]. With increasing N per leaf area, the proportion of total leaf N in the thylakoids remains the same as the proportion in the soluble protein increases. The growth of plants under lower irradiance greatly increases the partitioning of N into chlorophyll and thylakoids, while the electron transport capacity per unit of chlorophyll declines [31]—the plants are able to optimise the allocation of N in order to preserve a balance between the Calvin cycle (i.e., Rubisco) and light-harvesting (i.e., chlorophyll) capabilities [34]. The acclimation to light has been shown to affect N allocation within leaves [35,36]. The relative foliar chlorophyll concentration tends to increase with decreasing growth irradiance, while the fraction of N in Rubisco usually declines with decreasing irradiance [31,37]. The mentioned effects could explain the change in the level of yields between the analysed periods and changes in the level of chlorophyll content. In our study, in both types of cultivation, the yields of plants were lower in autumn compared to those in the spring period: for the substrate, they were about 31.8%; but in the case of hydroponics, they were about 17.2%—significant differences were found only for an N level up to 70 mg·dm$^{-3}$. Similarly, the chlorophyll content was lower in autumn samples compared to that in spring samples.

With increasing N nutrition of sweet chili (*Capsicum annuum* cultivar California wonder) (from 65 ppm to 260 ppm), the leaf green colour of intensity (greenness) varied from pale green to dark green [38]. In our experiment, in both the spring and autumn seasons of the substrate experiment, the leaf greenness of the flowering Chinese cabbage increased with an increase in the nitrogen level (from N-50 to N-90/N-110), but the greenness was decreased when the nitrogen treatment was higher than N-110. Actually, the change in leaf greenness had a similar trend to the yield changing—firstly increasing to a peak then inhibited under a high level of nitrogen treatment. The greenness colour can be an indicator of the chlorophyll level in plants' leaves, and the nitrogen level has a close link with the chlorophyll content [39]. It was also found in the study by Ibrahim et al. [40] that the higher the N supply, the higher the chlorophyll content in *Labisia pumila*; this was an effect of the increasing N level in leaves and increasing net photosynthesis rate with the increased N fertilisation.

### 4.2. Plant Nutrient Status

The previous studies of other authors [41–43] stated that N fertiliser increased the content of K, Ca, Mg, Na, Zn, and Mn in aboveground tissues. Meanwhile, other studies found that the content of Ca, Na, Zn, and Mn in aboveground tissues decreased with the N gradient [41,42,44,45]. The plants' response could be varied by species or the intensity of nitrogen nutrition. Nitrogen (N) and potassium (K) are two important mineral nutrients in the regulation of leaf photosynthesis [46]. In our studies, the ratios between N and K were multidirectional. According to Mulder's chart, there was an antagonism between N, K, and Cu with simultaneous synergism with Mg and Mo. In our studies, there were positive interactions between N, K, and Ca and an antagonism effect for Mg and Cu (for substrate cultivation); in the case of hydroponics, the positive impact was for P, Ca, Fe, Zn, and Cu. Smoleń and Sady [47] claimed that fertilisation with N (in the form of $NH_4NO_3$) led to a significant increase in the content of Fe in red cabbage.

We also found a positive trend of correlation between N and Fe in our study—the determined r-Pearson's coefficients were significant in both types of cultivation. Rutkowska et al. [48]

pointed out that nitrogen fertilisation caused an increase in the content of iron, manganese, zinc, and copper in a soil solution, which could influence plants in the uptake of nutrients. In our studies, we found a positive effect of N nutrition on the status of Mn and Zn in both the cases of plants cultivated in the substrate and hydroponically.

*4.3. Antioxidant Activity and Total Phenolic Content*

Choy sum exhibits potent antioxidant activity. In this study, a significant effect of N fertiliser on the TEAC, DPPH, and FRAP values was shown by medium N levels in the substrate cultivation system and high N levels in the hydroponic cultivation system. There are conflicting reports regarding the effect of N fertilisation on the antioxidant activity of plants [49–51]. Tavarini at al. [50] found that a high nitrogen supply had a negative effect on the phenolic content in *Stevia redaudiana* leaves. In contrast, a high antioxidant activity of onion was revealed by the highest nitrogen concentration [51]. Finally, the effect of N fertilisation on the antioxidant activity of Jerusalem artichoke varied depending on the cultivar and the method used for the antioxidant activity measurements [49].

The antioxidant activity of choy sum may be the result of the phenolic content. In this study, the highest TPC of choy sum was reported in spring samples at medium levels of N fertiliser (mostly N-90). A high level of N nutrition (N-130) did not result in the further increase of phenolic content (which was shown also by the PCA). The plants fertilised with high N levels tended to increase their photosynthesis, which consequently enhanced their biomass [52], which may have influenced the phenolic content [50]. This effect could be explained by the nitrogen-induced inhibition of phenolic compounds as a result of competition between phenolics and proteins in their precursor L-phenylalanine, as was explained by Margna et al. [53] and Tavarini et al. [50]. Phenolics are secondary plant metabolites whose biosynthesis runs through the shikimate pathway [54]. The first stage in their biosynthesis, which includes shikimic acid formation, leads to the biosynthesis of the aromatic amino acid L-phenylalanine. In the second stage, the deamination of L-phenylalanine by phenylalanine ammonia lyase (PAL) led to the formation of cinnamic acid and the following phenylpropanoid pathway and a further p-coumaroyl-CoA metabolite, which gives rise to phenolic compounds [55]. L-phenylalanine, the aromatic amino acid, is also the precursor of proteins. Thus, the increased use of L-phenylalanine for protein biosynthesis could result in its reduced availability for phenolic biosynthesis. The effect can occur in plants under abundant nitrogen nutrition [53], which was shown in our study.

According to our best knowledge, until now no study has investigated the effect of N nutrition and planting date (season) in various cultivation system on the phenolic content of choy sum. Other findings suggested that the higher the level of N fertilisation, the higher the accumulation of active metabolites such as phenolics. This was proved in onion that was subjected to two levels of nitrogen (urea), namely 130 kg/ha and 260 kg/ha, although there were no statistically significant differences between the two treatments [51]. Differences in phenolic content depending on the nitrogen uptake by plants were reported previously as well. The concentration of phenolics in pac choi increased when the N availability for plants dropped [56]. However, this increase was associated with a considerable yield reduction, thus the data were inconclusive because it remained unclear to what extent the phenolic compounds might change in the case of comparable productivity. The production of phenolic compounds by *Labisia Pumila* Blume under low levels of N fertilisation (0–90 kg/ha) was also found by Ibrahim et al. [40]. The TPC content was correlated with the enhanced PAL activity, which was followed by a reduction in soluble protein biosynthesis under low nitrogen levels, which indicated a higher availability of L-phenylalanine under low nitrogen content that stimulated the production of secondary metabolites such as phenolics [40]. The effect of N fertilisation on phenolics has been also reported in Jerusalem artichoke tubers; it was found to vary with the type of cultivar [49]. In general, nitrogen fertilisation increased the phenolic content in plants compared to non-fertilised plants (N-0), but the increase in the N level from 80 kg/ha to 120 kg/ha increased the phenolic content significantly only in the Gute Gelbe cultivar. For two other cultivars, namely Albik and Rubik, a higher

level of nitrogen fertilisation did not significantly influence the phenolic concentration in tubers [49]. Tavarini et al. [50] reported a significant effect of nitrogen fertilisation and harvest time on the phenolic content of *Stevia reboudiana* leaves. Regarding phenolic content, the highest values were reported for medium N levels (50 kg/ha) followed by N 150 kg/ha and N 300 kg/ha, whereas the total flavonoids were the highest for N 150 kg/ha followed by N0 (non-fertilised). This was in line with our findings.

## 5. Conclusions

The aim of this study was to determine the effect of increasing the nitrogen nutrition on the yield and some biochemical properties of flowering Chinese cabbage grown in two varied cultivation systems (substrate and hydroponic) and seasons (spring and autumn). The planting date varied the plant nutrition: in the substrate, the content of N, K, Ca, Fe, Mn, and Na in leaves was higher in autumn compared to spring; the same tendencies were found in the cases of P, K, Ca, Fe, Zn, Cu, Na in the hydroponic system. Nitrogen nutrition positively impacted plant yields—an effect that generally showed the same pattern in both seasons but varied depending on the type of cultivation. For the mean of two seasons, the yield increased from N-50 to N-90, which was an optimal level. More intensive N nutrition (N-110 and N-130) did not significantly modify plant yields. Moreover, the phenolic compound content increased with increasing N levels up to the medium doses (N-90 and/or N-110). A high supply of N fertiliser (N-130) did not increase the phenolic compounds of choy sum compared to medium and low doses. Generally, the optimal N dose for most samples was a medium N level (N-90), which indicated that the manipulation of N fertilisation might be an effective tool to obtain plants with a high content of bioactive compounds. The effect of season on the phenolic content and the antioxidant activity was also significant. The highest values of the parameters were obtained for the substrate spring samples. Altogether, the results obtained in this study provide useful information on choy sum's response to various N levels in two different cultivation systems and seasons and indicated that agricultural management provides the possibility to increase the physicochemical properties of choy sum from the nutritional and phytochemical points of view.

**Supplementary Materials:** The following supporting information can be downloaded at: https://www.mdpi.com/article/10.3390/agronomy12112869/s1, Table S1: SD values for content of N, P, K, Ca, and Mg in leaves (% in d.m.). Table S2: SD values for content of Fe, Zn, Mn, Cu (in mg·kg$^{-1}$ d.m.), and Na (% in d.m.) in leaves. Table S3: SD values for colour parameters. Table S4: SD values for chlorophylls and carotenoids. Table S5: SD values for phenolic content and antioxidant activity. Table S6: The r-Pearson's coefficients for the correlation between variables for substrate samples (as an average over season). Table S7: The r-Pearson's coefficients for the correlation between variables for hydroponic samples (as an average over season).

**Author Contributions:** Conceptualization, T.K. and W.L.; methodology, T.K. and M.M.-P.; investigation, W.L.; formal analysis: M.M.-P., T.K. and W.L.; visualization, M.M.-P., T.K. and W.L.; writing—original draft preparation, T.K., M.M.-P. and W.L.; writing—review and editing, T.K. and M.M.-P.; supervision, T.K. and M.M.-P.; project administration, T.K.; funding acquisition, T.K. All authors have read and agreed to the published version of the manuscript.

**Funding:** This study was financed by the programme "Maintaining research potential", which was financed by the Ministry of Science and Higher Education. This publication also was co-funded within the framework of the Ministry of Science and Higher Education programme "Regional Initiative Excellence" in 2019–2022 (No. 005/RID/2018/19), with financing in the amount of PLN 1,200,000.

**Institutional Review Board Statement:** Not applicable.

**Informed Consent Statement:** Not applicable.

**Data Availability Statement:** All data generated or analysed during this study are available from the corresponding author upon reasonable request.

**Conflicts of Interest:** The authors declare no conflict of interest. The funding sponsors had no role in the design of the study; the collection, analyses, or interpretation of data; the writing of the manuscript; or the decision to publish the research results.

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
