# Peer review of "Effect of Nitrogen Nutrition and Planting Date on the Yield and Physicochemical Parameters of Flowering Chinese Cabbage"

_agronomy, doi:10.3390/agronomy12112869_

Round 1

Reviewer 1 Report

I find the study interesting, as it is a novel crop. Knowing how it responds to nitrogen fertilization and how it influences parameters that affect nutritional quality and compounds of interest to health, is worthy.  

However, you should improve your English, it should be seen by someone fluent in English and improve this part. That is a easy task.

The biggest job you have is to change the discussion, and focus it on discussing your results with similar results. For example from L385 to L404 that's not discussing, that's giving very general information and then restating your results and somehow that general information explains them. That is not discussion. To discuss is to say why you get the results that you get, for example, in hydroponics and in spring the N concentration does not affect the growth, I suppose that with an adequate nutrient replenishment by the hydroponic system, the plants with N50 have been able to maintain a growth rate similar to N130. All this is what you have to discuss, and focus the references with your specific case and not give such general references.

In the introduction it is also missing your working hypothesis. Why two different culture conditions? Which is the reason of that?

In material and methods there is information that is repeated (e.g. L94 to L98). And please refer terms properly, "nutrient concentration" refer to units of a mineral per unit of fresh or dry weight, and "nutrient content" refer to the the total amount of that nutrient in a part of the plant, leaves, stem, shoor or root.

Author Response

 The reviewer 1 remarks concerned the following items:

- The biggest job you have is to change the discussion, and focus it on discussing your results with similar results. For example from L385 to L404 that's not discussing, that's giving very general information and then restating your results and somehow that general information explains them. That is not discussion. To discuss is to say why you get the results that you get, for example, in hydroponics and in spring the N concentration does not affect the growth, I suppose that with an adequate nutrient replenishment by the hydroponic system, the plants with N50 have been able to maintain a growth rate similar to N130. All this is what you have to discuss, and focus the references with your specific case and not give such general references.

Thank you for your comments. We do not agree. |In our opinion description of results and discussion are sufficient. Mentioning part from L385 to L404 is, in our opinion, important to understand the phenomen.

- In the introduction it is also missing your working hypothesis. Why two different culture conditions? Which is the reason of that?

Our hypothesis was that independent of season of cutlivation and system of cultivation nitrogen nutrition improve plant yielding – we add this sentence to manuscript. So it was a reason of conducting 4 independent experiments.
- In material and methods there is information that is repeated (e.g. L94 to L98). And please refer terms properly, "nutrient concentration" refer to units of a mineral per unit of fresh or dry weight, and "nutrient content" refer to the the total amount of that nutrient in a part of the plant, leaves, stem, shoor or root.

We improved the MM according to suggestions. In the text, the two terms were used interchangeably. We have homogenous into „content”|

Reviewer 2 Report

Dear Authors,

I reviewed your article titled in (Effect of nitrogen nutrition and season of cultivation on the yield and physicochemical parameters of flowering Chinese cabbage (Brassica campestris L. ssp. chinensis var. utilis Tsen et Lee). Overall, the data presented here is valuable to those working in this field demonstrates the effectiveness of a relatively simple intervention that could be applied a wider scale especially in the field of fertilization and quality. However, revision of the English grammar, sentence structure and extensive editing by a native speaker of all parts needs to be undertaken before this is at all ready for publication. There are some other major points that should be addressed in the individual sections.

I was stopped the revision process at the M & M part until you consider the following major problem:

- You have to re-analysis your data as three-factors (N levels – Planting data – growing media) and their interactions. Without this new statistical analysis, your MS could be rejected.

Other comments are mention in attached pdf file plus her:

Title:

1- planting data is better than season of cultivation

2- it is better to delete it and put in keywords

Abstract (many other comments in pdf file): 

-  line 12: you have to mention a sentence in the first about the problem that you need to solve

- line 13: you have to convert to hectare

- line 16: what refers to !! increased or decreased

- line 17-19: you can merge them in one sentence if you have the same results in substance and hydroponics

-line 20: what does you mean?? you don’t mention this rate in the first sentence. You have to present all the rates that used.

- line 23-24: you don’t mention which rate is better and which season.

- line 25 (keywords): use words differ than title.

Introduction:  

- line 32:  delete: However, this vegetable is still not familiar in Europe.

- line 36: did you measure it in the plants? it is important to see the effect of your treatments in the accumulation of S.

- line 42: revise all MS and don’t use this word. Just you mention increase, decrease, or non-significant.

-line: 43: you can use N as an abbreviation

-line: 46-48: also you could have nitrate poisoning. You could mention the lethal dose or maximum concentration in the plants

-line 48-49: you can merge this sentence with previous one.

-line 52: You don’t mention any information about the effect of planting date and planting substances. You have to add this information before this section.

Materials and methods (many other comments in pdf file):

Line 61: you have two or four? please revised

Line 61: which type? plastic, glass or fiber?? Mention dimension of the green house

Line 63: which types? present the chemical and physical properties of this soil.

Line 63: present more information about this beat!!

Line 64: which type of hydroponics?? Provides some pictures.

Line 66: if possible, present some pictures showing your plants and greenhouse

Line 68: complete randomize design

Line 68: replicates not replications

Line 69: why you use different number of replicates in both systems if you will compare between them?

Line 69: describe your pots dimension, material, and color

Line 70: you mention before that your greenhouse is not heated. Here you mention that under control?? if you have controlled greenhouse, why you study the effect of date of planting?

Line 77: provide some pictures

Line 80: I think this is a low saline condition (2.2 x 640 = 1400 ppm). What do you think?

Line 82: Present a brief paragraph about the preparing of seedlings

Line 90: were harvested and weighted or weighted with roots??

Line 92: normally we use -80 why you use -18?

Line 94: very low temperature?

Line 98: I think two days with this low temperature will not reach the plant to full dry. We normally use 60-65 C for 3-5 days until weight constant??

Line 149-151: why you use three methods? you don’t have experiment about the different methods? if the results are the same, use the most accurate one.

Result:

there is a problem in the statically analysis. why you don’t consider growing media as a factor? why you separate them? you have to make analysis as three factors (N-Levels, seasons, and growing media). Otherwise, you cannot compare between the both media (substrate and hydroponics. Some other comments:

Figure 1: what does the vertical access parameter? (fresh weight or what?) Please mention that. Delete the sub-horizontal lines. Delete the mean column, it is not providing more information. Convert unit to be mg/g or 100 g fresh weight. Do that in all figures.

Line 197: Where is the interaction between date and N-Levels?? provides SE or SD for every value.

Line 200-201: unclear sentence. Please re-phrase.

Line 2017: you have to run a new statically analysis with three factors (N-levels, growing media, and seasons). The results should be difference.

Author Response

 The reviewer 2 remarks concerned the following items: 

- You have to re-analysis your data as three-factors (N levels – Planting data – growing media) and their interactions. Without this new statistical analysis, your MS could be rejected.

 I disagree with the reviewer's comment. Due to differing properties, including air-water or sorption (mineral wool is an inert substrate), some differences could be incorrectly indicated. In fact, the two growing systems should be considered separately; hence the 2-factor statistic is correct.

- Title:  planting data is better than season of cultivation

It was changed according to Reviewer suggestion and we add more keywords.

Line 12: you have to mention a sentence in the first about the problem that you need to solve

It was changed according to Reviewer suggestion. We have add one sentence.

- Line 13: you have to convert to hectare

Thank you for your comments. We do not agree. Studies were conducted in growing media (not directly in soil) – it is possible to convert per dose/ha – 0-20 cm = 2 mlns dm3 x d (1,5 kg/L) =  3 mln kgs of soil – but it is not neccesery. The other cultivation system was also hydroponic where it is impossible to convert.

- Line 16: what refers to !! increased or decreased

It was changed according to Reviewer suggestion. We have add one sentence.

- Line 17-19: you can merge them in one sentence if you have the same results in substance and hydroponics

It was changed according to Reviewer suggestion

- Line 20: what does you mean?? you don’t mention this rate in the first sentence. You have to present all the rates that used.

We have add the description in abstract.

- Line 25 (keywords): use words differ than title.

It was changed according to Reviewer suggestion

Introduction:  

- Line 32:  delete: However, this vegetable is still not familiar in Europe.

It was changed according to Reviewer suggestion. We have delete the sentence.

- Line 36: did you measure it in the plants? it is important to see the effect of your treatments in the accumulation of S.

Thank you for your opinion, but mineralisation proccess for macroelements were in the mixture of sulphuric acid (20 cm3) with hydrogen peroxide (30%, analytically pure) – so it impossible to determined the S content. We will determined that macroelement in the future studies focus on nitrogen nutrition.

- Line 42: revise all MS and don’t use this word. Just you mention increase, decrease, or non-significant.

Our intention was to point out the fact that it generally modifies qualitative aspects, just like quantitative ones. We have deleted the word in the abstract, and also in a few places in all the manuscripts.

- Line: 43: you can use N as an abbreviation

It was changed according to reviewer suggestions. We have added the abbreviation.

- Line: 46-48: also you could have nitrate poisoning. You could mention the lethal dose or maximum concentration in the plants

It was changed according to reviewer suggestions.

- Line 48-49: you can merge this sentence with previous one.

It was changed according to reviewer suggestions. We have merge the sentence with previous one.

- Line 52: You don’t mention any information about the effect of planting date and planting substances. You have to add this information before this section.

Thank you for your suggestion. Details of phenomenus of planting date in moderate climate is unknown so we decided to do not show generall information.

Materials and methods (many other comments in pdf file):

- Line 61: you have two or four? please revised

We conducted 2 independent experiments in spring time (for soil:peat substrate and hydroponic) and the same 2 independent experiments in autumn time.

- Line 61: which type? plastic, glass or fiber?? Mention dimension of the green house.

It was changed according to reviewer suggestions. We added this informations: complex glasshouse sector, with an area of 272 m2.

- Line 63: which types? present the chemical and physical properties of this soil.

It was changed according to reviewer suggestions. In our studies we have used loamy sand, bulk density of 1.60 g·cm-3, total porosity 39%. Add to the next paragraph.

- Line 63: present more information about this beat!!

It was changed according to reviewer suggestions. We add much more information about peat.

- Line 64: which type of hydroponics?? Provides some pictures.

It was changed according to reviewer suggestions. we add that it was system with recirculation of nutrient solution

- Line 66: if possible, present some pictures showing your plants and greenhouse

We have  put pictures showing plants in both seasons of cultivation

- Line 68: complete randomize design

It was changed according to reviewer suggestions.

- Line 68: replicates not replications

It was changed according to reviewer suggestions.

- Line 69: why you use different tchem of replicates in both systems if you will compare between tchem?

We do not compare the system of cultivation because of varied physo-chemical properties (f.e. sorption).

- Line 69: describe your pots dimension, material, and color

Thank you for your comments. The information is bellow in manuscript. Plants were grown in yellow plastic pots (V 5 L).

- Line 70: you mention before that your greenhouse is not heated. Here you mention that under control?? if you have controlled greenhouse, why you study the effect of date of planting?

The goal was to demonstrate differences due to the seasons, which are marked by varying light levels.

- Line 77: provide some pictures

It was changed according to reviewer suggestions. We will add pictures.

- Line 80: I think this is a low saLine condition (2.2 x 640 = 1400 ppm). What do you think?

Yes. It is, especially compare with f.e. tomato.

- Line 82: Present a brief paragraph about the preparing of seedlings

It was changed according to reviewer suggestions.

- Line 90: were harvested and weighted or weighted with roots??

It was weight only edible parts of plant.

- Line 92: normally we use -80 why you use -18?

It standard procedure we use in our laboratory. The samples were store in plastic containers under nitrogen atmosphere to reduce the possibility of oxygen reaction.

- Line 94: very low temperature? Line 98: I think two days with this low temperature will not reach the plant to full dry. We normally use 60-65 C for 3-5 days until weight constant??

Temperature of drying depends on plant material. To improve drying we make small parts of plants and then dry them.

- Line 149-151: why you use three methods? you don’t have experiment about the different methods? if the results are the same, use the most accurate one.

The antioxidant action of extracted compounds could be an effect of various mechanisms, including radical scavenging capacity (based on different radicals) or metal chelating properties (reducing abilities). Thus, there is impossible to indicate only one accurate method for the antioxidant activity measurements in food which is very complex matrix. The most commonly, the antioxidant activity is investigated using various methods in order to show a wide spectrum of the antioxidant activity [1].

The most commonly used methods for the measurements of the radical scavenging properties are ABTS and DPPH assays. Both assays are based on single electron transfer (SET), although the hydrogen atom donation could also occur, and are using stable free radicals whose reduction by the antioxidant could be easily monitored by the spectrophotometer at visible region [2–5]. Both radicals are not relevant to biological system (they are non-physiological radicals) [6]. ABTS•+ cation radicals are soluble in aqueous and organic solvents, hence both non-polar and polar samples can be assessed. These cation radicals can be used at different pH which is very useful when studying the effectiveness of various plant materials [6]. Unlike DPPH radicals dissolve in organic solvents and are sensitive to acidic pH (the acidic pH causes the decolourization of DPPH radical solution which makes the radical scavenging ability of the antioxidant unable to measure) [6]. Additionally, the results of DPPH assay could be underestimated when monitoring the properties of samples containing anthocyanins since the color interference could occur between these flavonoids and DPPH solution [6]. Metal chelating properties of the samples are often investigated by the FRAP (ferric reducing antioxidant power) assay in which the reduction of ferric iron (Fe3+) to ferrous iron (Fe2+) is monitored [7]. The assay is also based on SET and is pH-dependent [7,8].

Thus, according to the authors it is inaccurate to choose only one method for the antioxidant activity measurements which was suggested by the reviewer, since the complexity of food matrix required to monitore various mechanisms of the antioxidant activity.

  1. Muzolf-Panek, M.; Stuper-Szablewska, K. Comprehensive study on the antioxidant capacity and phenolic profiles of black seed and other spices and herbs : effect of solvent and time of extraction. J. Food Meas. Charact. 2021, 15, 4561–4574, doi:10.1007/s11694-021-01028-z.
  2. Sánchez-Moreno, C.; Larrauri, J. a.; Saura-calixto, F. A procedure to measure the antiradical efficienc y of polyphenols. J. Sci. Food Agric. 1998, 270, 270–276.
  3. Miller, N.J.; Rice-Evans, C.; Davies, M.J.; Gopinathan, V.; Milner, A. A novel method for measuring antioxidant capacity and its application to monitoring the antioxidant status in premature neonates. Clin. Sci. 1993, 84, 407–412, doi:10.1042/cs0840407.
  4. Ozgen, M.; Reese, R.N.; Tulio, A.Z.; Scheerens, J.C.; Miller, A.R. Modified 2,2-azino-bis-3-ethylbenzothiazoLine-6-sulfonic acid (ABTS) method to measure antioxidant capacity of selected small fruits and comparison to ferric reducing antioxidant power (FRAP) and 2,2′-diphenyl-1- picrylhydrazyl (DPPH) methods. J. Agric. Food Chem. 2006, 54, 1151–1157, doi:10.1021/jf051960d.
  5. Mishra, K.; Ojha, H.; Chaudhury, N.K. Estimation of antiradical properties of antioxidants using DPPH - assay: A critical review and results. Food Chem. 2012, 130, 1036–1043, doi:10.1016/j.foodchem.2011.07.127.
  6. Shalaby, E.A.; Shanab, S.M.M. Comparison of DPPH and ABTS assays for determining antioxidant potential of water and methanol extracts of Spirulina platensis. Indian J. Mar. Sci. 2013, 42, 556–564.
  7. Benzie, I.F.F.; Strain, J.J. The ferric reducing ability of plasma (FRAP) as a measure of “Antioxidant Power”: The FRAP assay. Anal. Biochem. 1996, 239, 70–76.
  8. Nikolic, J.S.; Mitic, V.D.; Stankov Jovanovic, V.P.; Dimitrijevic, M. V.; Stojanovic, G.S. Chemometric characterization of twenty three culinary herbs and spices according to antioxidant activity. J. Food Meas. Charact. 2019, 13, 2167–2176, doi:10.1007/s11694-019-00137-0.

Result:

- there is a problem in the statically analysis. why you don’t consider growing media as a factor? why you separate them? you have to make analysis as three factors (N-Levels, seasons, and growing media). Otherwise, you cannot compare between the both media (substrate and hydroponics.

 I disagree with the reviewer's comment. Due to differing properties, including air-water or sorption (mineral wool is an inert substrate), some differences could be incorrectly indicated. In fact, the two growing systems should be considered separately; hence the 2-factor statistic is correct.

Some other comments:

- Figure 1: what does the vertical access parameter? (fresh weight or what?) Please mention that. Delete the sub-horizontal Lines. Delete the mean column, it is not providing more information. Convert unit to be mg/g or 100 g fresh weight. Do that in all figures.

The figures are improved.

- Line 197: Where is the interaction between date and N-Levels?? provides SE or SD for every value.

The SD values were present for each nutrient/season. Interaction are in the table (in columns described as spring and autumn). I think posting SD for each result will make the table unreadable and difficult to interpret. Relevant results are homogeneous groups.

- Line 200-201: unclear sentence. Please re-phrase.

Thank you for your comments. It was improved.

- Line 217: you have to run a new statically analysis with three factors (N-levels, growing media, and seasons). The results should be difference.

I disagree with the reviewer's comment. Due to differing properties, including air-water or sorption (mineral wool is an inert substrate), some differences could be incorrectly indicated. In fact, the two growing systems should be considered separately; hence the 2-factor statistic is correct.

In the manuscript we have also improved some sentences.

Round 2

Reviewer 1 Report

Thank you for considering some of my suggestions. I think the article can be published in Agronomy.

Author Response

Dear Reviewer,

Thank you!

Best regards,
Tomasz Kleiber

Reviewer 2 Report

Dear authors,

Thank you for your improvement in the article. There are still some minor comments:

1- Figure one: the significant letter is confused with SE bars. Please revised.

2- Table 1 and all tables: Please insert the SE after every value.

3- line 232: what does “Sason” mean? I think you mean Season.

4- The pictures are nice. It is better to include the pictures inside the article.

All the best

Author Response

The reviewer 2 remarks concerned the following items:

Thank you for your comments.
1. Figure one: the significant letter is confused with SE bars. Please revised.

It was changed according to Reviewer comment. Since the tables contain a great deal of data, designated SD, we inserted in the supplementary. This way they are presented and the tables in the text are more readable.

2. Table 1 and all tables: Please insert the SE after every value.

It was changed according to Reviewer comment. Since the tables contain a lot of data, designated SD, we inserted in the supplementary. This way they are presented and the tables in the text are more readable.

3. line 232: what does “Sason” mean? I think you mean Season.

Thank you. It was changed according to Reviewer comment.

4. The pictures are nice. It is better to include the pictures inside the article.

It was changed according to Reviewer comment.